# COVID-19 Obesity: Differences in Infection Risk Perception, Obesity Stress, Depression, and Intention to Participate in Leisure Sports Based on Weight Change

**DOI:** 10.3390/healthcare11040526

**Published:** 2023-02-10

**Authors:** Chulhwan Choi, Kyung-Rok Oh, Mun-Gyu Jun

**Affiliations:** 1Department of Physical Education, Gachon University, 1342, Seongnam-daero, Sujeong-gu, Seongnam-si 13120, Republic of Korea; 2Department of Coaching, College of Physical Education, Kyung Hee University, Seocheon-dong 1, Giheung-gu, Yongin-si 17104, Republic of Korea

**Keywords:** COVID-19, infection risk perception, obesity stress, depression, participation intention

## Abstract

This study examined the impact of COVID-19 on individuals’ weight change and mental health by analyzing differences in risk perception, obesity, stress, depression, and intention to participate in leisure sports during the COVID-19 pandemic. Data were collected in the Republic of Korea between June and August 2022. This study included 374 individuals aged ≥ 20 years who regularly participated in leisure sports. A comparative analysis subdivided the participants into two groups based on weight changes during the pandemic: weight loss and maintenance (Group 1) and weight gain (Group 2). These formed the independent variable. The dependent variables were (a) infection risk perception, (b) obesity stress, (c) depression, and (d) intention to participate in sports. The results revealed statistically significant differences between the two groups in infection risk perception, obesity stress, and depression factors, but not in the intention to participate in sports. This study demonstrated the impact of COVID-19 on weight changes and mental health. These findings can guide future quarantine strategies to control new infectious diseases and policies to prevent obesity and stress.

## 1. Introduction

COVID-19 is a disease caused by respiratory infections and is classified as a statutory first-class new infectious disease [1]. COVID-19 was first reported at the end of December 2019 and has since spread rapidly worldwide. The World Health Organization (WHO) declared it a pandemic in March 2020 [2]. As of August 2021, COVID-19 has caused more than 156 million confirmed cases and 3 million deaths worldwide, and various social measures have been implemented in Korea to prevent the spread of the disease [3]. Preventive measures against common infection routes include social distancing, prohibition of gatherings, restrictions on workplace operations, limits on face-to-face interaction, and induced, encouraged, and sometimes forced changes in daily life. In addition, fear and concern regarding viral infections have resulted in voluntary social distancing. Vaccines have been developed and administered to the general public, and the treatment of infected individuals has progressed smoothly [4].

However, “COVID-19 obesity,” which results from changes in daily life patterns owing to intense social distancing and extreme fear of infection, may linger. Obesity causes several diseases due to excessive fat accumulation in the body. The rate of obesity has increased due to reduced physical activity and behavioral radius since the COVID-19 outbreak [5]. According to a 2020 survey in Korea [3], 32.9% of respondents reported that they were not doing enough physical activity due to COVID-19, and 42.1% reported that they had experienced weight gain. The rate of adult obesity increased from 30% in 2014 to 34.6% in 2018; physical activity decreased from 58.3% in 2014 to 47.6% in 2018; these problems were further exacerbated by COVID-19 [3]. The World Health Organization (WHO) [6] declared COVID-19 and obesity a pandemic and emphasized the importance of prevention and management. Therefore, participation in exercise in the post-COVID-19 era may improve mental health and reduce the effects of obesity.

COVID-19 infection, deaths, viral infection awareness, economic changes, changes in personal environment, and changes in daily behavior can affect human health [7]. Weight changes affect daily behavior and future health, and weight gain trends have been reported in several countries since the COVID-19 outbreak [8,9]. These trends affect existing national obesity management plans, which aim to minimize health risks and improve individuals’ health. It is necessary to consider the impact of COVID-19 on weight and mental health. Therefore, the purpose of this study was to analyze differences in risk perception, stress, depression, and intention to participate in leisure sports based on weight change during the COVID-19 outbreak. The results of this study will provide useful insights into identifying psychological changes caused by the pandemic and possible future interventions.

### 1.1. Infection Risk Perception

Infectious disease risk perception refers to an individual’s subjective assessment of the seriousness and sensitivity of the infection; individuals perceive the risk and experience negative emotions when faced with or exposed to the risk of infection [10,11]. A study on COVID-19 risk recognition was conducted on 7000 individuals in 10 countries (Germany, Italy, Spain, Sweden, the United Kingdom, Korea, Japan, Australia, Mexico, and the United States). The impact of COVID-19 on individuals, families, friends, country of residence, and the possibility of infection and health condition seriousness after infection were evaluated. In addition, knowledge, experiences, social situations, trust in the government, science and medical staff, political tendencies, and the efficacy of COVID-19 vaccines were investigated. The results indicated that men had a lower risk perception than women in all countries, and higher risk perception was associated with a higher degree of preventive behaviors [12]. Lower COVID-19 knowledge and education levels were associated with lower risk perception [13]. Studies on respiratory infectious disease risk recognition, mainly conducted on nurses caring for such patients, and the effect of respiratory infectious disease risk perception on the use of personal protective equipment were recently conducted [14,15,16]. However, research on the perception of infectious disease risks among the general public is insufficient.

### 1.2. Obesity Stress

The stress that most individuals experience in their daily lives threatens their well-being, resulting in a lack of mental resources or exposure to work and situations beyond their capacity [17]. Restrictions on movement and meetings due to COVID-19 have led to drastic lifestyle changes. Social isolation measures increase negative eating behavior and decrease exercise frequency, resulting in poor health, which could cause weight gain and obesity [18]. Moreover, obese individuals may be more vulnerable to COVID-19 [3]. It follows that the ongoing COVID-19 pandemic may affect individuals’ weight changes. Thus, it is necessary to maintain a healthy lifestyle after the COVID-19 outbreak to prevent negative weight changes.

### 1.3. Depression

Depression results from a negative perception of oneself and is an emotional disorder indicated by anxiety, a sense of failure, loss, lethargy, and worthlessness [19]. Depression can be an obstacle that prevents individuals from performing their social roles properly and can lead to suicide in extreme situations. According to a survey conducted in March 2021 [20], one in five individuals (23%) was found to be at risk for depression, a six-fold increase from 2018. The percentage of individuals who thought about suicide within a 2-week period was 16.3%, a 3.5-times increase from 2018 [20]. Fear of an unpredictable future during the COVID-19 pandemic raised levels of anxiety and depression for individuals with existing psychiatric conditions and healthy people alike [21]. In addition, psychological challenges have reached a severe level, and prolonged and chronic conditions can lead to suicide and addiction. Therefore, it is necessary to develop psychological support and interventions.

### 1.4. Intention to Participate in Leisure Sports

Intention refers to a potential and planned decision regarding whether to perform an act at the individual decision-making level [22]. Intention to participate in sports activities is the easiest predictor of participation [23]. However, there is a possibility of participation, even if individuals do not intend to participate, if they show a favorable attitude toward certain sports [24]. This means that, even when one shows a favorable attitude, one’s intention to participate in sports may be lowered for certain reasons. It has been proposed that even when people’s attitudes favor non-performance, they are more likely to lead to behavior when they are willing to participate [25]. Owing to the COVID-19 pandemic, an increasing number of individuals have changed their types of exercise to simple muscle exercises and stretches that can be performed at home rather than in public sports facilities or gymnasiums [26]. In addition, many leisure sports participants experienced moderate abandonment and decreased motivation to exercise due to COVID-19 [27]. However, owing to changes in social perception after the COVID-19 pandemic, leisure sports with a small number of individuals and home training increased compared to activities in sports facilities. In addition, due to the spread of non-contact culture, face-to-face sports programs have been converted to non-face-to-face online classes [28].

## 2. Materials and Methods

### 2.1. Survey Participants and Data Collection Procedure

This study analyzed the differences in infection risk perception, obesity stress, depression, and intention to participate in sports depending on individuals’ weight changes during the COVID-19 pandemic but after the global easing of COVID-19 quarantine. Participants in the survey were adults over 20 years of age in the Republic of Korea who regularly participated in leisure sports. Data were collected from July to August 2022 using the convenience sampling method (a non-providence sampling method) and was applied to comply with the exact qualifications of the survey participants. A quantitative survey was then conducted. An online survey platform provided by Google was used for data collection. All survey respondents were informed of the purpose of this study and voluntarily participated using the self-administration method. A total of 374 questionnaires were collected.

Basic demographic information (e.g., sex and age) was collected, and additional information about weight from before and after the COVID-19 pandemic was obtained. The survey participants were divided into two groups based on weight change before and after the pandemic: a weight loss or maintenance group (Group 1) and a weight gain group (Group 2). Specifically, participants who did not gain more than 3% of their initial weight were defined as having lost or maintained weight, and those who increased more than 3% of their initial weight were defined as having gained weight. The standard (±3% weight change) for classifying groups followed a study of Orsama, Mattila, Ermes, van Gils, Wansink, and Korhonen [29], which analyzed the weight changing by period. Weight change was used as an independent variable for the comparative analysis.

The main survey questionnaires for comparing and analyzing the mental health of the two groups, subdivided according to weight change, were as follows (20 items): (a) infection risk perception, (b) obesity stress, (c) depression, and (d) intention to participate in sports. The research design focused on analyzing weight change and mental health, which have emerged as social problems due to the COVID-19 pandemic. Table 1 presents the demographic information of the survey respondents.

### 2.2. Instrument

The tool used to measure respondents’ risk perception of respiratory infectious diseases, such as COVID-19, was applied to a questionnaire modified by Lee [15], which is based on the risk perception tool for acute infectious respiratory diseases by Kang and Kim [14]. The risk perception measurement tool for infectious respiratory diseases consists of two sub-factors: (a) sensitivity (three items) and (b) seriousness (four items). The tool used to measure depression was developed by Kee and Lee [30] and modified by Lee [31]. The depression measurement tool consisted of four items comprising of a single factor. In addition, the tool used to measure obesity stress was applied using a questionnaire revised by Ryu [32] based on the Body Attitudes Questionnaire (BAQ) developed by Ben-Tovim and Walker [33]. The obesity stress measurement tool consists of two sub-factors: (a) body satisfaction (three items) and (b) comparison with others (three items). The tool used to measure the intention to participate in sports after experiencing the COVID-19 pandemic was derived from previous studies [34,35,36,37]. The tool consists of three items comprising of single factor. The questionnaire used in this survey used a five-point Likert scale ranging from 1 (“not at all”) to 5 (“very much”).

### 2.3. Data Analysis

The collected data were analyzed using SPSS version 23.0. First, the analysis showed descriptive statistics for the survey respondents, including sociodemographic data (e.g., sex, age, type of household, income level, and favorite sports). Second, to verify the scale validity of the data collected, confirmatory factor analysis (CFA) was performed for four factors: infection risk perception, depression, obesity stress, and intention to participate in sports). Third, Cronbach’s alpha was used to test the reliability of the data. Finally, multivariate analysis of variance (MANOVA) was conducted to investigate the differences in dependent variables between the two groups segmented by an independent variable (i.e., weigh change).

## 3. Results

### 3.1. Scale Validity and Reliability

First, CFA was performed to determine scale validity using six variables, including sub-factors. All statistical results from the CFA met statistical standards. Additionally, the goodness-of-fit test reported acceptable results (CMIN = 383.280, DF = 151, CMIN/DF = 2.538, NFI = 0.900, CFI = 0.936, and RMSEA = 0.064). Second, Cronbach’s alpha was used to determine the reliability of the survey items based on a statistical cut-off value of 0.70 [38]: sensitivity (infection risk perception), α = 0.807; seriousness (infection risk perception), α = 0.769; depression, α = 0.913; body satisfaction (obesity stress), α = 0.810; comparison with others (obesity stress), α = 0.708; and intention to participate in sports, α = 0.808. Therefore, all the instruments used in this study demonstrated satisfactory statistical reliability. Table 2 shows the detailed results of the CFA and Cronbach’s alpha.

### 3.2. MANOVA

MANOVA was used to determine the differences in infection risk perception, depression, obesity stress, and intention to participate in sports (Table 3). First, the homogeneity of covariance was tested (Box’s *M* = 57.936, *F* = 2.711, *p* < 0.05). Statistically significant differences were observed between the two groups (Wilks’ lambda = 0.837, *F* = 11.924, *p* < 0.05, partial *η*^2^ = 0.163). Specifically, statistically significant mean differences were found regarding obesity stress in (a) sensitivity to infection risk perception, (b) seriousness of infection risk perception, (c) depression, (d) body satisfaction with obesity stress, and (e) comparison with others. The survey respondents in Group 1 reported higher mean scores for the five factors than those in Group 2. However, the two groups had no significant difference in the intention to participate in sports. Table 4 presents the detailed mean scores of the six dependent variables for the two groups.

## 4. Discussion

The spread of COVID-19 had an unprecedented impact on the world, and the resulting changes have tremendously affected individuals’ lives. Quarantine policies and social distancing were severely restrictive, and many individuals were psychologically affected by fear of infection [39]. Although vaccine development and vaccinations have progressed since 2021, fear of infection has yet to dissipate, while social and economic impacts have persisted. Efforts have been made to address the social damage caused by the COVID-19 outbreak and restore a state of relative normalcy. At the personal level, weight gain due to reduced physical activity associated with COVID-19 prevention measures has emerged as a social problem [40]. Excessive weight gain and obesity have long been regarded as negative factors that affect physical and mental health. Therefore, this study analyzed perceptions of individuals’ psychological health. This included infection risk perception, depression, obesity stress, and participation in physical activities since the start of the COVID-19 pandemic.

Group 2 showed high sensitivity and seriousness—two sub-factors of infection risk perception. Previous studies revealed a close relationship between weight gain and lifestyle [41,42,43]. Furthermore, COVID-19 has significantly impacted individuals’ life patterns, either arbitrarily or unintentionally [39]. Specifically, social distancing limits social relationships and isolated individuals, resulting in limited physical activities, including going out. Consequently, physical activity in a limited space using online content has emerged as the next best option [44,45]. However, this second-best option was insufficient to bring individuals’ lives back to pre-COVID-19 levels. Ultimately, it is inferred that individuals’ psychological sensitivity and the seriousness of COVID-19 infection forced sedentary life patterns, resulting in weight gain. In addition, the highest average value among all factors examined in this study was for infection risk perception. This could explain the current situation in which confirmed infection cases continue to occur despite the development of vaccines.

Furthermore, Group 2 showed significantly higher results for depression than Group 1. According to the WHO [46], depression is a major cause of disability and a large contributor to the world’s diseases. Depression can have fatal adverse effects on maintaining a healthy life, as it can last long or recur. Previous studies have reported that depression, a fatal mental illness that can occur in various social situations, can be caused by weight gain [47,48]. In particular, depression and weight gain are closely related, as the influence of each factor appears regardless of the cause and effect; that is, depression can lead to weight gain [48,49]. It could be inferred that during the COVID-19 outbreak, individuals experienced social isolation and limited physical activity, resulting in weight gain and depression. Generally, physical activity generates physical, mental, and social benefits [50]. The current situation, whereby the COVID-19 quarantine policy was relaxed for mental health, might be the best time to address individuals’ mental health. Currently, the number of confirmed cases has decreased, and the number of cured patients has increased; therefore, urgent attention should be paid to weight gain and depression, which are social problems caused by COVID-19.

Moreover, obesity stress was significantly different between the two groups. According to the Seoul National University Hospital [51], BMI—height divided by the square of the body weight—defines overweight as 25 kg/m^2^ or more and obesity as 30 kg/m^2^ or more. However, in everyday life, it is common for individuals to subjectively judge overweight and obesity based on their own perceived weight gain rather than on medical figures. Therefore, satisfaction with weight is subjective and relative [52]. Although data related to weight changes before and after the COVID-19 outbreak were collected, this study focused on the changes in weight that individuals subjectively felt rather than on medical figures to analyze the psychological factors of stress. This analysis revealed that Group 2 experienced higher stress than Group 1. In addition, both types of stress experienced through the evaluation of the participant’s body and comparison with others were analyzed. The results showed that individuals were stressed regarding their bodies, which had changed due to weight gain; moreover, they experienced more stress than others. Although appropriate stress can help motivate individuals [53], previous studies on obesity and stress reported that stress negatively affects an individual’s mental health [54]. Therefore, it is necessary to resume physical activity in the current situation. Intention to participate in physical activity was high in both groups, indicating that individuals were willing to live healthy lives through physical activity. The results of this study can be used to guide meaningful changes and interventions for weight loss.

## 5. Conclusions

The potential for new infectious diseases to occur at any time, even after COVID-19, has caused anxiety and concern worldwide. Therefore, this study analyzed the impact of weight change by comparing and analyzing individuals’ weight changes and mental health during the COVID-19 pandemic. This study focused on the differences in infection risk perception, obesity, stress, depression, and intention to participate in sports based on weight change before and after the pandemic: a weight loss or maintenance group (Group 1) and a weight gain group (Group 2). As a result, Group 2 showed more negative results for infection risk perception, depression, and obesity stress factors than Group 1. This result shows that the government’s social distancing policy and quarantine implemented during the COVID-19 outbreak limited people’s physical activities, and the resulting weight gain had a psychological impact on the public.

## 6. Research Limitations

In spite of the meaningful implications of the results, this study had practical research limitations. This study found that the overweight population increased after the outbreak of COVID-19, and as a result, it had a negative impact on the mental health of the public. However, due to the lack of previous studies that analyzed weight changes and mental health after the COVID-19 outbreak, it may be ambiguous whether the results of this study are influenced by the passage of time or changes due to the pandemic. In addition, depression is not only affected by the factors of obesity stress, but it can also occur for a variety of other reasons, such as the COVID-19 pandemic situation itself or changes in mental status. Therefore, future studies should find a clearer link between COVID-19 obesity and mental health through a long-term research design.

Next, the mental health that an individual can experience is very complex and diverse. Moreover, fear and social control caused by the coronavirus, which has not been experienced before, must have had a significant impact on the psychological health of the public. However, this study conducted quantitative research through questionnaires, so the public’s thoughts can be inferred, but there may be limitations in analyzing the mental state of a complex individual in depth. Therefore, in the future, more specific research should be conducted through qualitative research methods.

Finally, a limitation of this study is that the outbreak of COVID-19 was not predicted. In other words, in order to analyze weight change and mental health, objective data before the outbreak of COVID-19 should be based. However, due to the unpredictable outbreak of COVID-19, as mentioned above, this study was conducted in response through a survey rather than past data. However, this may be a realistic research limitation of most studies comparing before and after the outbreak of COVID-19. Nevertheless, the research attempt of this study is of great significance in that it prepares for the outbreak of viruses that may occur again in the future.

## Figures and Tables

**Table 1 healthcare-11-00526-t001:** Descriptive statistics.

		Group 1	Group 2
		Weight Loss or Maintenance Group	Weight Gain Group
Sex	Male Female	122 (58.1%) 88 (41.9%)	65 (39.6%) 99 (60.4%)
Age	20s 30s 40s Over 50	51 (24.3%) 45 (21.4%) 52 (24.8%) 62 (29.5%)	43 (26.2%) 48 (29.3%) 41 (25.0%) 32 (19.5%)
Type of household	Single-person Two or more people	31 (14.8%) 179 (85.2%)	26 (15.9%) 138 (84.1%)
Monthly income level	Less than USD 1000 USD 1000–3000 USD 3000–5000 USD 5000–7000 More than USD 7000	37 (17.6%) 64 (30.5%) 61 (29.0%) 30 (14.3%) 18 (8.6%)	25 (15.2%) 50 (30.5%) 46 (28.0%) 31 (18.9%) 12 (7.3%)
Participation in sports	Indoor sports Outdoor sports Home training	51 (24.3%) 101 (48.1%) 58 (27.6%)	40 (24.4%) 60 (36.6%) 64 (39.0%)
Total		210 (100.0%)	164 (100/0%)

**Table 2 healthcare-11-00526-t002:** Results of CFA (validity) and Cronbach’s alpha (reliability).

Construct and Scale Items	*λ*	AVE	C.R.	*α*
**Infection Risk Perception**				
**(Sensitivity)** I feel at risk of being infected with COVID-19. I am worried that I might be infected with COVID-19. I feel there is a high possibility of being infected with COVID-19.	0.710 0.831 0.680	0.553	0.786	0.807
**(Seriousness)** If I got infected with COVID-19, my family would get infected. My life could be in danger if I got infected with COVID-19 by participating in sports. If I got infected with COVID-19 by participating in sports, my daily life would become uncomfortable. If I got infected with COVID-19 by participating in sports, I would have to get intensive treatment.	0.679 0.709 0.690 0.803	0.521	0. 812	0.769
**Depression** I feel discouraged and depressed because of my weight. My life is worthless because of my weight. I feel disturbed by trivial things because of my weight. I feel like crying because of my weight.	0.830 0.859 0.864 0.857	0.727	0.914	0.913
**Obesity stress**				
**(Body satisfaction)** I am not satisfied with my weight at the moment. I need to control my weight. I have no confidence in my life because of my figure.	0.659 0.642 0.821	0.507	0.753	0.810
**(Comparison with others)** A person with a nice figure looks attractive. When I see fit people, I compare them to my body. When a friend watches their weight, I also pay attention.	0.519 0.774 0.730	0.467	0.719	0.708
**Intention to participate in sports** I will continue to participate in sports activities. I will recommend sports activities to people around me. I will purchase equipment for participation in sports.	0.838 0.767 0.702	0.594	0.814	0.808

Note. CMIN = 383.280, DF = 151, CMIN/DF = 2.538, NFI = 0.900, CFI = 0.936, RMSEA = 0.064, *λ* = Factor loading, AVE = Average Variance Extracted, C.R. = Composite Reliability, *α* = Cronbach’s alpha coefficient for reliability.

**Table 3 healthcare-11-00526-t003:** Results of MANOVA.

Variables	Sub-Factors	*df*	*F*	*p^a^*	*η^2^*
Infection risk perception	Sensitivity Seriousness	1 1	8.630 8.389	0.004 ** 0.004 **	0.023 0.022
Depression		1	50.372	0.000 ***	0.119
Obesity stress	Body satisfaction Comparison with others	1 1	53.893 17.433	0.000 *** 0.000 ***	0.127 0.045
Intention to participate in sports		1	0.083	0.775	0.000

Note. *** *p* < 0.001, ** *p* < 0.01.

**Table 4 healthcare-11-00526-t004:** Mean scores of six dependent variables for each group.

	Infection Risk Perception		Depression	Obesity Stress		Intention to Participate in Sports
	Sensitivity	Seriousness		Body Satisfaction	Comparison with Others	
Group 1	3.75	3.75	2.22	3.25	3.36	3.54
Group 2	4.01	**3.97**	**2.97**	**3.94**	**3.72**	3.51

Note. Group 1 = weight loss or maintenance group, Group 2 = weight gain group; significantly higher mean scores among the groups are shown in bold.

## Data Availability

Not applicable.

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
