# Peer review of "COVID-19 Obesity: Differences in Infection Risk Perception, Obesity Stress, Depression, and Intention to Participate in Leisure Sports Based on Weight Change"

_healthcare, 2023, doi:10.3390/healthcare11040526_

Round 1

Reviewer 1 Report

The article is scientifically very interesting, however, it has some minor aspects that need to be corrected.

Introduction. The introduction described the well state of the art of study problem, but this section needs some small changes.

From lines 41 to 68 it describes “reducing physical activity and obesity and depression”, I think in a somewhat repetitive way. In addition to the fact that there are other sections with the same content below. I think you should unify it.

Material and methods. The experiment of this study is well designed.

Results. The results and discussion are described the findings of the manuscript, and the section sign up to date literature which closely related to the study topic.

Authors should discuss the limitation of the study, apart from the conclusions.

Bibliography. Most of references listed in the manuscript is up to date, although few are not recent.

In some references the year is in bold, not in others. References must be from the last 5 years.

Reference 8,9,12. The year does not appear.

Reference 24,26 are not recent.

Author Response

Reviewer 1

The article is scientifically very interesting, however, it has some minor aspects that need to be corrected. Introduction. The introduction described the well state of the art of study problem, but this section needs some small changes. Based on your comments, the manuscript has been revised.

From lines 41 to 68 it describes “reducing physical activity and obesity and depression”, I think in a somewhat repetitive way. In addition to the fact that there are other sections with the same content below. I think you should unify it. We definitely understand your concern. The section of Introduction has been modified by deleting the duplicate contents.

Material and methods. The experiment of this study is well designed. We really appreciate to your favorable review.

Results. The results and discussion are described the findings of the manuscript, and the section sign up to date literature which closely related to the study topic. We really appreciate to your favorable review.

Authors should discuss the limitation of the study, apart from the conclusions. Based on your comment, the section of 6. Research limitations has been discussed (apart from the section of 5. Conclusions).

Bibliography. Most of references listed in the manuscript is up to date, although few are not recent. According to your comments, the section of References has been revised as many as possible. (Recent references have been added.)

In some references the year is in bold, not in others. References must be from the last 5 years. Based on the template provided from the journal (Healthcare), manuscript formatting has been revised additionally. Also, recent references have been added.

Reference 8,9,12. The year does not appear. The reference #8 and 9 have been deleted for this revision process. The reference #12 cited from the electronic source (Website) does not necessarily contain the year according to the template provided from the journal (Healthcare).

Reference 24,26 are not recent. The reference # 24 and 26 have been replaced.

Reviewer 2 Report

1The structure of the "introduction" is not logical, where the purpose  ,meaning should be put at the end of the “introduction”.

2 In the "introduction", the reason why the research was performed is not stated clearly. In another word, why you compare the difference between risk perception, stress, depression, and intention to participate in leisure sports based on weight change during the COVID-19 outbreak.

3 The research compared the difference between two groups,  which is a crosssectional study, while how could you control the impact of  two groups' baseline scores of risk perception, stress, depression, and intention to participate in leisure sports.

4 From the line of 152 to 158 , the grouping method was described, could you explain why you grouped the participants according to the 3% standard?

5 In188-190 lines,  the independent  variable of multivariate analysis of variance should be described.

6 in Talbe 2 , λ,AVE,C.R.,andΑ should be annotated .

7 According to the results of MANOVA,  are there interaction effect between the independent variable?

8 In the part of conclusion,  the logic is not clear , especially in the "limitation". In addition, clinical meaning of the research result should be supplemented.

Author Response

Reviewer 2

The structure of the "introduction" is not logical, where the purpose, meaning should be put at the end of the “introduction”. In the "introduction", the reason why the research was performed is not stated clearly. In another word, why you compare the difference between risk perception, stress, depression, and intention to participate in leisure sports based on weight change during the COVID-19 outbreak. Based on your comment, the section of Introduction has been revised. In addition, the purpose of this study has been put at the end of Introduction.

The research compared the difference between two groups, which is a cross-sectional study, while how could you control the impact of two groups' baseline scores of risk perception, stress, depression, and intention to participate in leisure sports. We totally understand your concern. All research in social science focusing on rapid changing society with quantitative research design might have practical research limitations. As you mentioned, this research also has had a limitation. Thus, the research limitation which could not control all factors has been added at the end of the Research limitations. If further research is conducted in the future, all factors will be controlled as much as possible.

From the line of 152 to 158, the grouping method was described, could you explain why you grouped the participants according to the 3% standard? To maintain maximum objectivity, the standard (3%) used in the cited literature were applied in this study. The reference has been definitely added.

In188-190 lines, the independent variable of multivariate analysis of variance should be described. Added.

in Table 2, λ,AVE,C.R. and Α should be annotated. Added.

According to the results of MANOVA, are there interaction effect between the independent variable? We totally understand your concern. However, this study implemented the one-way MANOVA with an independent variable (weight change) and more than two dependent variables (Infection risk perception, Depression, Obesity stress, and Intention to participate in sports). The interaction effect between the independent variable is computed from two-way MANOVA design. Thus, the interaction effect could not be calculated.

In the part of conclusion, the logic is not clear, especially in the "limitation". In addition, clinical meaning of the research result should be supplemented. The sections of the Conclusions and Research limitations have been revised. 

Round 2

Reviewer 2 Report

After the revision, It is better!